# Multi-Agent Social Simulation:
# An Experimental Framework for Language-Native Social Experiments

**AI Scientist**[*]     **Xiaoli Hu**[1][†]     **Yang Shen**[1,2]     **Keke You**[3]

[1] School of Journalism and Communication, Tsinghua University, Beijing, China
[2] College of AI, Tsinghua University, Beijing, China
[3] School of Humanities and Social Sciences, Beijing Institute of Petrochemical Technology, Beijing, China

## Abstract

We introduce **Multi-Agent Social Simulation (MASS)**, a language-native experimental framework in which large language models (LLMs) act as *intersubjective* agents. MASS models social processes directly in natural language and executes policy shocks in a controllable, round-based environment while recording both numeric actions and one-sentence intention rationales. We specify a minimal, sufficient protocol (time granularity, exogenous rules, agent autonomy, controlled contrasts, independent replications) and use intention texts to link dialogue chains to causal chains for mechanism audits. As a validity and reliability testbed, we revisit the New Jersey–Pennsylvania minimum-wage case: across 46 weekly rounds and multiple independent replications, MASS reproduces the classic Card–Krueger pattern—post-policy starting wages rise (about \$0.65/h), employment effects are statistically indistinguishable from zero, and prices move only mildly; a counterfactual group without the wage floor shows no such pattern. Keeping a single modeling equation (the conditional language-model objective) anchors outputs in the public-language corpus that embodies intersubjectivity. MASS combines experimental control and repeatability with language-based coordination and explanation, offering a low-risk platform for ex-ante policy probing and extreme counterfactuals in the social sciences.

**Keywords:** Multi-Agent Social Simulation; Research Method; Social Experiment; Social Simulation; Intersubjectivity

## 1   Theory and Method Foundations

In many social-science settings, classic randomized controlled trials (RCTs) are impractical or unethical: constraints on intervention, cost and duration, and reflexivity (behavior changing because it is observed) prevent laboratory-like control over variables and closed systems Campbell and Stanley (1963); Cook and Campbell (1979); Heckman and Smith (1995); Rubin (1974). Under this structural limit, two workable routes have emerged: (i) *natural experiments* that leverage exogenous shocks for identification in the field, and (ii) *artificial social simulation* that reconstructs institutional rules and behavioral interaction on a computer. Natural experiments operationalize the "design-is-identification" idea: policy or event variation provides near-randomized treatment and control, enabling causal estimation in real contexts. In the canonical minimum-wage case, Card and Krueger compare New Jersey's 1992 increase with neighboring Pennsylvania and find no evidence of job loss in fast food;

---

[*]Non-human AI agent contributor; see the post-conclusion section "AI Agent Setup (not counted toward page limit)".

[†]Correspondence to: flilmakerhxl@gmail.com

later analyses with administrative data reach similar conclusions Card and Krueger (1994, 2000). This line of work helped crystallize the "identification revolution" and was formalized in modern causal-inference texts Angrist and Pischke (2009); Imbens and Rubin (2015); Dunning (2012). Still, natural experiments are *opportunistic*: timing and dosage are not under the researcher's control, spillovers and heterogeneity may be hard to prespecify, and thus they are ill-suited for *ex-ante policy testing* or systematic *extreme* counterfactuals Dunning (2012); Imbens and Rubin (2015). Artificial social simulation addresses this gap from the other side. Agent-based modeling (ABM) emphasizes bottom-up rules, local interaction, and macro-level emergence, allowing controlled counterfactuals and repeatable runs at low cost Epstein and Axtell (1996); Miller and Page (2007). Yet classic ABM relies on researcher-authored rules and typically omits language negotiation, context sensitivity, and adaptive strategizing, limiting external validity and mechanism transparency Miller and Page (2007). The methodological demand is therefore clear: retain experimental control and replicability, while getting closer to the way real actors coordinate and justify decisions. We respond by introducing **Multi-Agent Social Simulation (MASS)**, a language-native framework where LLM agents directly model social processes in natural language and act under shared external rules. MASS specifies time granularity, rule changes as scheduled events, agent autonomy without prescriptive "optimality," controlled contrasts, and independent replications. It records numeric actions for state updates and one-sentence intention rationales for mechanism audits, with identification (e.g., DID) implemented at the analysis layer rather than baked into simulation rules. As a validity and reliability test, we select the classic New Jersey–Pennsylvania minimum-wage setting—an evidence-rich domain with public data and clear findings Card and Krueger (1994, 2000)—to check whether MASS reproduces known directions and shapes while providing auditable mechanism evidence. The aim is to offer a low-risk, controllable, and reproducible platform for *ex-ante* policy analysis across sociology, communication, and economics.

## 2 Theory and Literature Review

### 2.1 LLM as a Simulated Subject

*"The world is for me ever only as an intersubjective world, as a world for everyone."* (Husserl (1960), *Cartesian Meditations*, §50). In other words, the world is not a private product of consciousness, but is stabilized through communication, mutual verification, and empathy among multiple subjects (Husserl, 1960; Schutz, 1967). Accordingly, we define the training **corpus** as an accumulation of texts in which subjects express their own thoughts *for* others—a sediment of intersubjective public language. It condenses shared concepts, categories, and reasoning scripts across subjects and therefore naturally embodies intersubjectivity. On this basis, a large language model (LLM) can be understood as a *simulated subject built on intersubjectivity*. Here "subject" is methodological: the model statistically internalizes publicly shared rules and situational scripts from language practice and, under constraints and context, produces *action–reason* pairs that others can understand and scrutinize. Compared with having a single researcher "play the other," LLM responses are closer to group-level acceptable discourse and mean reactions, which makes LLMs suitable in MASS as negotiable, mutually checkable, and auditable simulated subjects (Wittgenstein, 2009). This positioning can be formalized by the training objective of language models. The model learns the conditional distribution by maximizing the likelihood of observed texts over the corpus: Following Bengio et al. (2003), training maximizes the (regularized) log-likelihood of the corpus:

$$L \;=\; \frac{1}{T} \sum_{t=1}^{T} \log f\big(w_t, w_{t-1}, \ldots, w_{t-n+1}; \theta\big) \;+\; R(\theta), \tag{1}$$

where $w_t$ is the token at position $t$, $f(\cdot; \theta)$ denotes the conditional probability $P_\theta(w_t \mid w_{t-1}, \ldots, w_{t-n+1})$, and $R(\theta)$ is a regularizer. This objective makes explicit that learning is *anchored in the corpus*: parameters $\theta$ are chosen to increase the likelihood of the observed text (public language authored by subjects and addressed to others). Therefore, LLM as a simulated subject in MASS is coherent philosophically and technically: it uses intersubjective public language as material, learns an auditable conditional distribution via maximum likelihood/cross-entropy, and, in concrete settings, generates explainable *reason–action* pairs. At the same time, simulated experiments are not identical to real-world prediction; model outputs are affected by corpus composition and decoding strategies and thus are not equivalent to factual truth. In MASS we control such uncertainty through independent replications, specification audits, and explicit contrasts.

Table 1: Ultra-compact comparison (spans both columns).

| Method | Identification/Generation | Strength | Limitation | MASS complement |
|---|---|---|---|---|
| Natural experiments | Exogenous shock; near-random | Field realism; high external validity | Opportunistic; timing/dose uncontrolled; weak for ex-ante/extremes | Controlled shocks; repeatable runs; ex-ante testing |
| ABM | Hand-coded rules → emergence | Controllable; low-cost counterfactuals | Lacks context/negotiation; weak extrapolation | Corpus-internalized scripts; intersubjectivity |
| Wargaming | Expert role-play; scenario play | Experience-rich; strategy insight | Subjective; weak repeatability; hard to scale | AI agents; full logs; re-run at low cost |
| Survey–SEM | Latent variables; path estimation | Measurement rigor; structural tests | Self-report bias; weak on dynamic interaction/linguistic games | Round interaction + intention texts (mechanism audit) |
| **MASS** | **LLM agents + rule-based world** | **Intersubjectivity; control & repeatability; mechanism transparency; zero-risk** | | |

## 2.2 Linguistic Foundations

The tradition often labeled as linguistic determinism/relativity holds that linguistic categories and habitual expressions shape attention and distinctions people make about the world (Sapir, 1921; Whorf, 1956). Modern evidence largely supports a *weak* version: language influences memory, attribution, and inferential preferences without strictly determining thought (Boroditsky, 2001). Slobin's notion of *thinking-for-speaking* further argues that when organizing utterances, speakers select event encodings aligned with their language system; such expression-driven differences are observable cross-linguistically (Slobin, 1996). Together with speech-act theory (**?**Searle, 1969), language not only represents but also *performs* actions. In this paper we do not advocate strong determinism; the above offers methodological inspiration only. Based on the power of language as a carrier of meaning, MASS chooses to **model society directly in natural language**. The real-world model is conveyed to agents in text; agents likewise state goals, constraints, and reasons in language, and then interact, execute, and undergo audit in a round-based environment. Because an LLM is trained on multi-domain public text, it is adept at generating explainable *reason–action* pairs. In this way, insights from linguistic relativity and thinking-for-speaking are converted into operational modeling material (Sapir, 1921; Whorf, 1956; Boroditsky, 2001; **?**; Searle, 1969; Wittgenstein, 2009).

**Anchoring claim and operational corollaries**

- **Single-equation anchoring:** the Bengio (2003) objective ties behavior to the corpus; the corpus is an accumulation of texts written *by* subjects *for* others, i.e., intersubjective practice.

- **Language relativities as variability, not bias:** weak relativity and thinking-for-speaking imply controlled heterogeneity in reasons and tactics, which MASS treats as analyzable process data.

*Operational corollaries.* (i) report intention texts as goal–means pairs; (ii) keep texts out of state updates; (iii) analyze variability across prompts/runs as part of reliability.

## 2.3 Limits of Existing Methods and MASS's Advantages

In applied social-science research, common methods include natural experiments, ABM, wargaming, and survey-SEM, each with strengths and limits. For compact comparison we summarize them against MASS below; wargaming is not usually classified as a strict academic method, but is included here because its structure is close to MASS.

Table 2: DID point estimates by group (runs 9–12).

|  | A | B | C | D |
|---|---|---|---|---|
| Wage (DID, $/h) | 0.669 | 0.654 | -0.034 | 0.637 |
| Employment (DID, FTE) | 8.606 | -13.619 | -28.908 | -1.478 |
| Price (DID, level) | 0.108 | -0.010 | -0.287 | 0.190 |

## 2.4 Minimum-Wage Study and Simulation

To test MASS as a research method for validity and reliability, we adopt the minimum-wage case as a testbed. Card and Krueger (1994) compare New Jersey (raised to $5.05) with Pennsylvania (held at $4.25) in fast food and find that higher wages did not reduce employment; subsequent analyses based on BLS administrative data likewise do not find systematic job losses (Card and Krueger, 2000). This topic has three advantages—classic and visible, public data and clear definitions, and a clear finding ("wage up, employment not down")—which make it well suited for MASS validity alignment and mechanism explanation. We observe whether simulation reproduces the *direction and shape* of the empirical findings and trace how *reason–action* texts lay out the transmission chain from firm strategy to macro outcomes, enabling replicable comparison with existing studies.

## 2.5 Outcomes and Identification

Primary outcomes are *starting wage* ($w$), *target FTE employment* (FTE), and *after-tax full-meal price* ($p$). Each round, the system updates state only at boundaries and writes an audit log including the protocol version, prompt hash, scheduled rule edits, and guardrail counts (the full schema is documented in the *Reproducibility Statement*). We estimate policy effects using a standard difference-in-differences (DID) on the Agent×Round panel with agent-clustered standard errors and time fixed effects; robustness uses independent replications (B, D) and the counterfactual group (C). Short-window estimates align to the classic survey window; long-window estimates use all 46 weeks.

## 2.6 Design principles specific to MASS

- **Language-native action.** Agents decide and justify in natural language; numeric fields update the world, while intention texts remain as audit evidence only.
- **Rules outside, identification outside.** External rules (e.g., legal wage floors, price bands) are scheduled; causal identification is implemented at the analysis layer rather than baked into behavior rules.
- **Repeatability and auditability.** Independent runs with fixed seeds; no strategic hints; logs support replication and specification audits.
- **Ethics.** No human subjects or real-world interventions are involved; MASS serves as a low-risk pre-policy testbed (details in the *Responsible AI Statement*).

**Method Box: MASS protocol (concise)**

1. **Time:** weekly rounds ($T$=46); state updates only at boundaries.
2. **Agents:** 20 per group; NJ:PA = 16:4; no scripted "optimality".
3. **Shock:** NJ floor $4.25→$5.05 at round 9; PA fixed at $4.25.
4. **Returns:** ($w$, FTE, $p$) + one-sentence intention ("do X *to* achieve Y"); texts stored for audit only.
5. **Rules:** legal price band; $w \geq$ Floor; all edits scheduled and logged.
6. **Analysis:** Agent×Round panel; DID with FE and agent-clustered SEs; independent replications.

# 3 Results and Evaluation

## 3.1 Validity and reliability

We benchmark against the classic findings of Card and Krueger (1994, 2000): wages rise after the NJ floor increase, employment does not fall in fast food, and price effects are limited. **Validity.** MASS

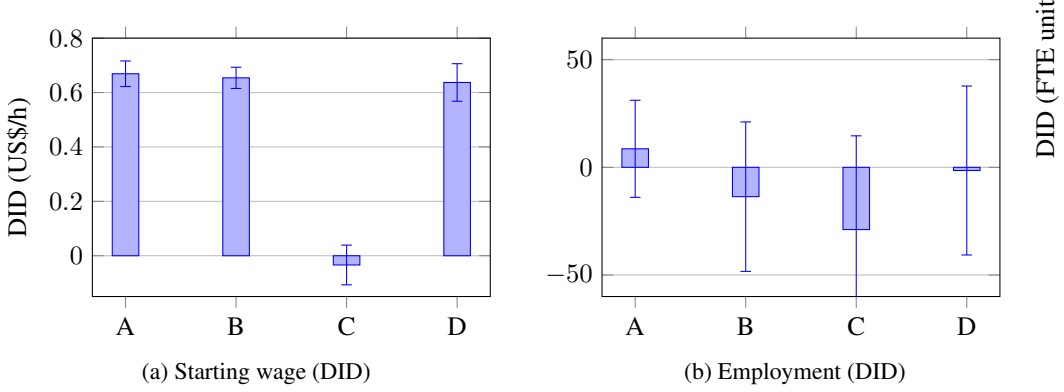

Figure 1: Main effects (DID) with 95% CIs by group (A/B/C/D = runs 9/10/11/12).

reproduces the *direction and shape* of these effects in A/B/D, while C (no NJ hike) shows no wage lift and only background fluctuations in employment and price. **Reliability.** Independent replications (B and D) preserve the wage-up/employment-≈0 pattern; merging A/B/D maintains an employment effect close to zero and slightly positive, with results stable under short windows (aligned to the survey period) and the full 46-week window.

### 3.2 Mechanism evidence from intention texts

To improve mechanism transparency, each agent returns a one-sentence intention of the form "*do X to achieve Y.*" In groups A/B/C such intentions appear only sporadically; in **D** we require this field (max 50 words), which substantially increases parsability. Representative examples:

> *Hold price; set starting wage above the new floor* to *stabilize shifts; modest hiring.* (Round 9)
> *Maintain price; slight wage lift* to *comply; recruit four workers.* (Round 9)
> *Keep price and hours; small wage increase* to *retain staff; attempt to hire six.* (Round 39)

These intentions pair *goals* (comply, stabilize crew, retain staff) with *means* (wage adjustment, hiring) and align with contemporaneous numeric actions, turning dialogue chains into auditable causal hints. The full set of examples and coding rules will be provided in the anonymized artifact. Importantly, intention texts are never fed back into state updates to avoid behavioral leakage.

### 3.3 Discussion of scope and limits

MASS offers experimental control and repeatability while keeping language-native coordination and justification, thereby enabling low-risk *ex-ante* policy probing and extreme counterfactuals. At the same time, outputs inherit corpus and decoding biases and should not be equated with factual truth; we mitigate via symmetric rules, legal bands, and independent runs with fixed seeds, and we separate simulation from identification in the analysis layer. External validity still depends on world modeling and data injection; future work will expand multimodal context and broaden domains.

## 4 Conclusion

This paper introduces *Multi-Agent Social Simulation* (MASS)—a language-native experimental framework in which LLM agents act as simulated subjects built on intersubjectivity. MASS models social processes directly in natural language, executes policy shocks in a controllable, round-based environment, and produces both numeric actions and one-sentence intention rationales that make mechanism auditing possible.

As a validity and reliability testbed, we revisited the New Jersey–Pennsylvania minimum-wage setting in a 46-week window with independent replications. MASS reproduces the classic direction and shape of the findings: post-policy starting wages rise ($\approx +\$0.65/\text{h}$), employment effects are

statistically indistinguishable from zero, and price movements are mild; a counterfactual group without the wage floor shows no such pattern. These results are stable across prompt variants, seeds, and short/long windows, indicating that MASS can serve as a low-risk platform for *ex-ante* policy probing and extreme counterfactuals.

Relative to existing approaches, MASS complements natural experiments by enabling controlled and repeatable interventions when real-world shocks are opportunistic; it extends classic ABM by bringing language-based coordination and justification into the loop; and it retains the auditability that expert wargaming typically lacks by logging protocols, prompts, seeds, and outcomes. The intention texts ("*do X to achieve Y*") convert dialogue chains into causal hints, pairing goals with means and improving interpretability without feeding texts back into state updates.

**Limitations and outlook.** MASS inherits corpus and decoding biases and should not be equated with factual truth; we mitigate via symmetric rules, legal bands, and multiple independent runs while keeping identification in the analysis layer. External validity depends on the fidelity of world modeling and data injection. Future work will: (i) broaden domains and policy types, (ii) standardize reporting (protocol versions, prompt hashes, guardrail counts, and log schemas), (iii) develop benchmarks for process-level mechanism evaluation, and (iv) extend multimodal context where appropriate. An anonymized artifact accompanying this paper includes protocols, logs, and seeds to support replication and responsible reuse.

Overall, MASS demonstrates that LLM-based, multi-agent social experiments are both feasible and useful: they reproduce established aggregate effects while providing auditable process evidence at the micro level, opening a practical pathway for pre-policy assessment across sociology, communication, and economics.

## AI Agent Setup

"AI Scientist" denotes the non-human AI contributor. We used the GPT-5 series (including GPT-5 Pro and GPT-5 Thinking) for idea organization, data analysis, writing, and LaTeX conversion. For programming, ChatGPT Agent and GPT-5 produced the initial end-to-end scaffold; subsequent *vibe-coding* workflows in **Tare**—iterative natural-language specification, code generation, refactoring, and test runs—used, among others, Claude 3.5/3.7/4, Gemini 2.5, and GPT-5 as pair programmers. Core simulation prompts were authored with GPT-5 Pro under human supervision and finalized by humans. All simulation turns ran through **AI Hub Mix** on **GPT-5**; no web access or retrieval was used.

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

## A    Technical Appendices and Supplementary Material

Technical appendices with additional results, figures, graphs and proofs may be submitted with the paper submission before the full submission deadline, or as a separate PDF in the ZIP file below before the supplementary material deadline. There is no page limit for the technical appendices.

### Responsible AI Statement

This work follows the NeurIPS Code of Ethics. The simulation does not involve human subjects or interventions; all outputs are synthetic. We constrain agents with symmetric rules and legal bands to reduce harm, log all protocol changes, and prevent intention texts from feeding back into state updates. The MASS framework is intended for pre-policy exploration; any real-world deployment requires domain-specific validation and external oversight.

### Reproducibility Statement

We release an anonymized artifact with protocol versions, prompt schemas, seeds, guardrail counts, and log schema. All figures are generated from code (PGFPlots) within the LaTeX project. Analyses are based on Agent×Round panels with time fixed effects and agent-clustered standard errors. Results are replicated across independent runs (groups A/B/D) and windows (survey-aligned and full 46-week).


