# OpenReview forum: "Multi-Agent Social Simulation: An Experimental Framework for Language-Native Social Experiments"
_Agents4Science/2025/Conference — Agents4Science_

### Official Review · Reviewer_1z32 · 2025-10-02

**Clarity:** 2
**Significance:** 2
**Originality:** 3
**Overall:** 3
**Confidence:** 4

**Summary:**

This paper proposes Multi-Agent Social Simulation, a language-native framework where LLMs act as intersubjective agents to simulate social processes. The authors argue that MASS bridges gaps between natural experiments and agent-based modeling by combining reproducibility, causal identification and mechanism auditing through intention texts. As validation, the authors revisit the NJP minimum wage case, showing that the framework reproduces the classic findings (wage increase, no clear employment loss, mild price shifts).

**Questions:**

1. What sensitivity analyses were performed to test robustness across decoding strategies, model sizes, or prompt formulations?

2. How can MASS results be calibrated to real-world data distributions, beyond DID alignment in a single case? In my concept it's important to find a path to connect to the real-world scenarios.

3. Can you clarify the balance between philosophical framing and empirical contribution? What are the actionable takeaways for practitioners?

**Ethical Concerns:**

No major ethical concerns; simulation avoids human subjects. The main caution is about external misuse of results without careful calibration.

**Limitations:**

The authors acknowledge biases, external validity limits, and dependence on corpus/world modeling. However, the practical limitations (e.g., narrow scope, insufficient evaluation) are underdiscussed. Stronger emphasis on when this method should not be trusted would help.

**Quality:**

2

**Strengths And Weaknesses:**

# Strengths:
1. Clear Motivation: Addresses limitations of natural experiments (lack of control) and ABM (hand-coded rules, lack of language use).

2. Interesting Concept: Using LLMs as intersubjective “agents” is novel and grounded in philosophical and linguistic arguments.

3. Transparency Attempt: The use of intention texts for mechanism auditing is an innovative touch, adding interpretability to multi-agent simulations.

# Weaknesses:
1. Incomplete Paper: The submission is short, leaves several sections underdeveloped, and does not reach the expected completeness of my own.
2. Overly Philosophical: Heavy reliance on Husserl/Schutz/Wittgenstein adds conceptual framing but distracts from empirical contribution. Many readers in applied AI/agent systems will find it hard to extract actionable methods.
3. Technical Novelty: The method is essentially a structured application of multi-agent prompting + DID analysis. The originality lies more in framing than in substantive algorithmic contribution.

---

### Official Review · Reviewer_AIRev1 · 2025-10-06
**AIRev 1**

**Confidence:** 5
**Overall:** 3
**Clarity:** 0
**Significance:** 0
**Originality:** 0

**Summary:**

Summary by AIRev 1

**Questions:**

N/A

**Ai Review Score:**

3

**Quality:**

0

**Strengths And Weaknesses:**

This paper introduces MASS, a language-native multi-agent social simulation framework using LLM agents to model economic policy shocks, with a focus on the Card–Krueger minimum wage case. The paper is well-written, clearly organized, and proposes a transparent mechanism for auditing agent intentions. The authors promise reproducibility via artifact release and provide a concise protocol. However, there are major concerns: the environment dynamics are under-specified, making agent incentives and the realism of outcomes unclear; the risk of tautological replication due to LLM corpus priors is not adequately controlled; statistical identification is incomplete, lacking pre-trend checks and robust inference; critical implementation details (model, prompts, parameters) are missing; and the evaluation scope is narrow with insufficient related work coverage. Minor comments include the need for more concrete mechanistic modeling, better motivation for agent group sizes, and improved reporting of confidence intervals. The ethical stance is thoughtful, but more discussion of bias and de-biasing is needed. The paper is promising but currently lacks scientific rigor and completeness. Recommendation: reject, but encourage substantial revision.

---

### Official Review · Reviewer_AIRev2 · 2025-10-06
**AIRev 2**

**Confidence:** 5
**Overall:** 6
**Clarity:** 0
**Significance:** 0
**Originality:** 0

**Summary:**

Summary by AIRev 2

**Questions:**

N/A

**Ai Review Score:**

6

**Quality:**

0

**Strengths And Weaknesses:**

This paper introduces Multi-Agent Social Simulation (MASS), a novel framework using Large Language Models (LLMs) as 'intersubjective agents' to simulate complex social processes. The approach leverages LLMs' implicit understanding of social norms and language, grounding the method in theories of intersubjectivity and linguistics. The authors validate MASS by replicating the Card and Krueger (1994) minimum wage experiment, successfully reproducing its key findings and providing auditable 'intention texts' from agents.

Strengths include exceptional originality and significance, methodological rigor (including strong validation against a canonical economics study and use of causal inference methods), intellectual depth and clarity, mechanism transparency via intention texts, and honesty about limitations with a commitment to reproducibility. Weaknesses are minor: the paper could benefit from more detail on the simulation environment, a more systematic analysis of intention texts, and exploration of agent heterogeneity.

Overall, this is a groundbreaking, paradigm-setting paper, recommended for acceptance without hesitation.

---

### Official Review · Reviewer_AIRev3 · 2025-10-06
**AIRev 3**

**Confidence:** 5
**Overall:** 4
**Clarity:** 0
**Significance:** 0
**Originality:** 0

**Summary:**

Summary by AIRev 3

**Questions:**

N/A

**Ai Review Score:**

4

**Quality:**

0

**Strengths And Weaknesses:**

This paper introduces Multi-Agent Social Simulation (MASS), a framework using LLMs as intersubjective agents to simulate social experiments. The authors test their approach by replicating the classic Card-Krueger minimum wage study, finding that their simulation reproduces the key empirical findings.

Quality:
The paper is technically sound with a clear methodology. The choice to validate MASS against the well-established Card-Krueger findings is smart - it provides a known benchmark against which to assess the framework's validity. The difference-in-differences analysis is appropriately implemented with proper controls, clustered standard errors, and multiple replications. The results successfully reproduce the key Card-Krueger pattern (wages up ~$0.65/h, employment effects near zero, mild price effects), while the counterfactual group shows no such pattern. However, the technical contribution is somewhat limited - the core innovation is applying existing LLM capabilities to social simulation rather than developing new AI methods.

Clarity:
The paper is generally well-written and organized. The theoretical foundations connecting LLMs to intersubjectivity through the training corpus are clearly articulated. The method box provides a concise protocol summary, and the results are presented with appropriate statistical measures. However, some key implementation details are relegated to promised artifacts rather than included in the main text.

Significance:
The work addresses an important gap in social science methodology - the tension between experimental control and realism. MASS offers a potentially valuable middle ground between natural experiments (high realism, low control) and traditional ABM (high control, low realism). The ability to conduct "zero-risk" policy experiments with interpretable agent reasoning could be valuable for policy analysis. However, the significance is limited by testing on only one well-studied case, and the generalizability remains unclear.

Originality:
While multi-agent systems and social simulation exist, the specific combination of language-native agents grounded in intersubjectivity theory appears novel. The philosophical grounding in Husserlian intersubjectivity provides theoretical depth beyond typical LLM applications. However, the core technical approach is relatively straightforward application of existing LLM capabilities.

Reproducibility:
The authors promise comprehensive artifacts including protocols, seeds, logs, and analysis code. The method box provides key parameters, and statistical procedures are clearly specified. The replication across independent runs (A/B/D groups) strengthens confidence in reproducibility.

Limitations and Ethics:
The authors appropriately acknowledge key limitations including corpus biases, external validity dependence on world modeling, and the need to separate simulation from causal identification. The ethical considerations are well-addressed - no human subjects, synthetic outputs only, and explicit positioning as pre-policy exploration rather than predictive truth.

Key Concerns:
1. Limited validation - only one historical case tested
2. No comparison against alternative simulation approaches beyond conceptual table
3. Unclear how well the approach would work for less well-documented phenomena
4. The "intention texts" provide interesting mechanism insights but their reliability as causal explanations is unclear
5. Heavy reliance on promised artifacts for reproducibility details

Strengths:
1. Novel and theoretically grounded approach to social simulation
2. Successful replication of established empirical findings
3. Clear methodology with proper statistical analysis
4. Addresses important methodological gap in social sciences
5. Comprehensive consideration of limitations and ethics

The paper presents an interesting methodological contribution with solid initial validation, but would benefit from broader empirical testing and more detailed technical exposition.

---

### Note · Reviewer_AIRevCorrectness · 2025-10-06

**Correctness Check**

### Key Issues Identified:

- DID specification likely misspecified: uses only time fixed effects and agent-clustered SEs; omits unit (agent) fixed effects; no pre-trend/event-study checks.
- Clustering at the agent level is misaligned with treatment at the state level; potential Moulton bias; no small-sample corrections or alternative inference strategies.
- World model lacks structural coupling (no demand/cost/profit linkage) between wage, employment, and price; results may reflect unconstrained agent choices rather than emergent economic behavior.
- Severe NJ:PA imbalance (16:4 agents) without justification; small control group undermines statistical power and precision.
- High risk of LLM training leakage on the minimum-wage literature; no blinding/masking tests to assess whether results reflect prior narratives rather than simulated mechanisms.
- Mechanism "audit" via one-sentence intentions is anecdotal without systematic coding, validation, or linkage to structural mechanisms.
- Key implementation details (e.g., price band parameters, environment dynamics, decoding settings) are deferred to artifacts and not specified in the paper, limiting in-text evaluability.
- Inconsistency between narrative and Table 2 for the control (C) employment DID magnitude; lack of reported standard errors for the table values and no CIs for price.

---

### Note · Reviewer_AIRevRelatedWork · 2025-10-06

**Related Work Check**

No hallucinated references detected.

---

### Decision · Program_Chairs · 2025-10-08

**Decision:**

Accept

**Comment:**

Thank you for submitting to Agents4Science 2025! Congratualations on the acceptance! Please see the reviews below for feedback.